# Design of a Multi-Node Data Acquisition System for Logging-While-Drilling Acoustic Logging Instruments Based on FPGA

**DOI:** 10.3390/s25030808

**Published:** 2025-01-29

**Authors:** Zhenyu Qin, Junqiang Lu, Baiyong Men, Shijie Wei, Jiakang Pan

**Affiliations:** 1National Key Laboratory of Petroleum Resources and Engineering, China University of Petroleum (Beijing), Beijing 102249, China; qinzy_cupb@163.com (Z.Q.); bymen@cup.edu.cn (B.M.); w1696848448@163.com (S.W.); cup888888@163.com (J.P.); 2Beijing Key Laboratory of Earth Prospecting and Information Technology, China University of Petroleum (Beijing), Beijing 102249, China

**Keywords:** logging-while-drilling acoustic logging instrument, FPGA, acquisition system, AD7380

## Abstract

The logging-while-drilling (LWD) acoustic logging instrument is pivotal in unconventional oil and gas exploration, and in providing real-time assessments of subsurface formations. The acquisition system, a core component of the LWD acoustic logging suite, is tasked with capturing, transmitting, and processing acoustic signals from the formation, which directly affects the accuracy and timeliness of the logging data. Recognizing the constraints of current LWD acquisition systems, including limited data collection capabilities and inadequate precision, this study introduces an FPGA-based multi-node data acquisition system for LWD acoustic logging. This system increases sampling density and data accuracy, leading to a more comprehensive collection of formation information. The multi-node acquisition system is composed primarily of a main control circuit board and several acquisition circuit boards, all connected via an RS485 bus. The Field-Programmable Gate Array (FPGA) is utilized to develop the acquisition circuit board’s firmware, offering adjustable control over parameters, such as the AD7380’s operational mode, sampling rate, and depth, facilitating real-time and concurrent acquisition and storage of formation acoustic signals. The main control board communicates with the acquisition boards via the RS485 bus, issuing commands to enable autonomous data collection and transfer from each board, thus enhancing the system’s reliability and scalability. Experimental results confirm the system’s capacity to efficiently capture waveform signals and upload them in real-time, underscoring its dependability and timeliness. The findings suggest that the system is capable of high-speed, real-time acquisition and processing of acoustic signals, offering robust technical support for the continued application of LWD acoustic logging instruments.

## 1. Introduction

In the realm of oil and gas exploration, there is a growing imperative for precise evaluation of subsurface rock formations and the efficient development of hydrocarbon reservoirs. Within the domain of acoustic logging, logging-while-drilling (LWD) acoustic logging instruments have emerged as indispensable tools in geological prospecting and drilling operations, owing to their real-time and high-precision capabilities [1,2,3,4,5,6]. These instruments are poised to supplant traditional wireline logging projects, and their sphere of application is ever-expanding. Prominent oilfield service providers globally are actively engaged in the research and enhancement of next-generation LWD acoustic logging instruments. Notable examples include Schlumberger’s SonicScope and Baker Hughes’ APX instruments. The SonicScope instrument is equipped with an advanced wideband emission transducer and a 4 × 12 array of receiving transducers, which significantly enhances measurement accuracy and resolution. It features a variety of emission modes, including monopolar and quadrupolar excitation. The APX instrument employs active transducer technology, with a receiving assembly comprising a 4 × 6 array of transducers, enabling the reception of multipole acoustic data, and thereby improving the success and efficiency of oil and gas exploration endeavors. The design of the acquisition system within LWD acoustic logging instruments is crucial for obtaining comprehensive subsurface information [7,8,9,10,11,12,13,14]. At present, in the field of acoustic logging, the large-scale array azimuth remote detection instrument has a good application prospect. As more and more information is collected and the imaging function is more and more perfected, the acquisition system that receives multiple receiving transducers becomes more and more important. Therefore, the research of multi-node acquisition systems based on FPGA development can effectively adapt to the current oil industry environment, and better collect more detailed formation information. However, traditional LWD acoustic logging instruments are constrained by limitations, such as restricted data acquisition capabilities, slow transmission rates, and inadequate sampling densities. These constraints also manifest in data processing and real-time performance, hindering the effective acquisition of valuable subsurface rock-layer information under complex geological conditions [15]. To surmount these challenges, this study presents the design and implementation of an FPGA-based multi-node acquisition system for LWD acoustic logging instruments capable of supporting an 8 × 8 array of receiving transducers. The system adopts a modular, top-down design paradigm, leveraging the high-speed processing and parallel processing capabilities of an FPGA to perform real-time acquisition and processing of acoustic signals. This approach effectively enhances the velocity and efficiency of data acquisition and processing, enabling real-time and precise capture of formation acoustic signals, and thereby providing more reliable technical support for geological exploration.

## 2. System Overview

The logging-while-drilling acoustic logging instrument is primarily composed of a main control circuit, multiple acquisition nodes, and excitation electronic circuits. The main control circuit serves as the central hub of the instrument, tasked with managing and coordinating the entire system’s operations. It receives and processes data from the telemetry section, dispatches control signals to the excitation circuit via the instrument communication bus, and communicates with each acquisition node to collect and preliminarily process the acoustic signals from subsurface formations. The excitation circuit primarily receives control signals from the main control circuit and generates the necessary excitation signals accordingly. The acquisition circuit is chiefly responsible for gathering signals from the subsurface and relaying them to the main control circuit through the instrument communication bus. Figure 1 provides a schematic diagram of the electronic system of the LWD acoustic logging instrument.

The main control circuit board is connected to the acquisition circuit board and the excitation circuit board via a bus, collectively forming the electronic system of the logging-while-drilling acoustic logging instrument. When the equipment commences operation, the telemetry system dispatches device instructions and other information to the LWD acoustic logging instrument. The main control circuit board begins to parse the commands, extracting valid ones, and reframes the commands for broadcast dissemination via the bus. The acquisition circuits and excitation circuits within the LWD acoustic logging instrument initiate corresponding configurations based on the received commands. Subsequently, they transmit data back in sequence according to the directives from the main control board, enabling the analysis of subsurface information and reflective interfaces.

The FPGA-based multi-node acquisition system, as depicted in Figure 2, is designed with a modular architecture that includes a main control circuit board and up to eight analog acquisition boards. The system’s flexibility is enhanced by allowing the number of analog acquisition boards to be adjusted to meet specific requirements, thus improving its scalability and adaptability. The main control board interfaces with each analog acquisition board via an RS485 bus. It features an integrated Digital Signal Processor (DSP) + FPGA architecture, where the DSP is tasked with issuing telemetry commands and forwarding the collected data, while the FPGA oversees communication among the acquisition nodes on the RS485 bus, managing data transmission, storage, processing, and the dispatch of broadcast commands. The analog acquisition boards, upon receiving the broadcast commands, configure the Analog-to-Digital Converter (ADC) accordingly, adjust the gain, and store the acquired data in a First-In-First-Out (FIFO) buffer that is ready for retrieval by the main control board. This design enables the system to accommodate a variable number of acquisition nodes through the RS485 bus, making it suitable for applications of varying sizes. By employing FPGA technology for data acquisition, command parsing, and bus control, the system achieves high scalability, real-time performance, and reliability.

The main control circuit board primarily employs a DSP + FPGA architecture. Within the DSP, there are several key modules: the network interface module, the bus communication module, and the algorithm module. The network interface module is responsible for communication with the telemetry system, while the bus communication module facilitates communication between the DSP and the FPGA on the main control board. The algorithm module is tasked with analyzing, filtering, and computing interface waves from the acquired subsurface data. The FPGA within the main control board includes the bus interface module, data storage module, and command parsing module. The bus interface module handles communication between the DSP and the FPGA as well as communication with the acquisition and excitation circuit boards. The data storage module is responsible for storing the collected data and adding frame numbers to the data uploaded from different acquisition boards. The command parsing module receives command information from the DSP, decodes the commands, and reframes them for RS485 bus communication.

The acquisition board, due to its requirement to collect a large amount of data with high real-time requirements, utilizes the FPGA as the main control chip. The FPGA on the acquisition board includes the data acquisition module, data communication module, and command parsing module. The data acquisition module is responsible for controlling the ADC chip, conducting data collection, and programmatically setting the gain of the PGA chip. The data communication module is in charge of communication with the FPGA on the main control board, including receiving commands and uploading collected data. The command parsing module decodes commands issued by the FPGA on the main control board, extracting key information such as sampling depth, sampling interval, sampling delay, and gain amplification factors.

The main control board incorporates the TMS320F28377D, a high-performance DSP from TI. This DSP is endowed with an internal 32 KB Static Random-Access Memory (SRAM) for data buffering, and is replete with an array of peripheral interfaces and formidable processing capabilities. Its dual-core architecture enables each core to operate at speeds up to 200 MHz, and it supports external memory interfaces for Synchronous Dynamic Random-Access Memory (SDRAM) as well as a Serial Communication Interface (SCI) for serial communication. These features adeptly fulfill the requirements of the present design endeavor. The FPGA selected is the XC7K325T from AMD, which delivers an impressive 326,080 logic cells, 840 DSP slices, and a memory capacity of 16,020 KB. It also offers up to 500 I/O pins, meeting intricate interface demands, and is equipped with substantial embedded memory and distributed RAM, providing ample storage resources for a diverse range of applications.

The FPGA utilized on the acquisition board is the 10M16SCU169I7G, known for its abundant logic elements, robust clock management capabilities, diverse memory interfaces, and advanced security features, which hold broad application prospects in the oil logging field. Its power supply requirement of only 3.3 V simplifies hardware design and alleviates the design pressure on the power board. The rich I/O pins can meet the simultaneous control requirements for data acquisition across eight channels. The ADC chip selected is the AD7380 from Analog Devices, a high-performance 16-bit/14-bit analog-to-digital converter renowned for its dual-channel synchronous sampling, fully differential analog input, up to 4 MSPS throughput rate, and up to 92.5 dB signal-to-noise ratio. It integrates an on-chip oversampling function to enhance dynamic range and reduce noise, features resolution enhancement capabilities, includes a built-in 2.5 V reference voltage source, and supports high-speed SPI interfaces, operating over a temperature range of −40 °C to +125 °C, making it suitable for various industrial and data acquisition applications.

The instrumentation amplifier employed is the AD8429 from Analog Devices, an ultra-low noise instrumentation amplifier characterized by its extremely low input noise, up to 90 dB common-mode rejection ratio (CMRR), and precise DC performance. It supports a wide supply voltage range, with gain adjustable from 1 to 10,000 via a single resistor, features a bandwidth of 15 MHz, and a slew rate of 22 V/μs. Operating over an industrial temperature range of −40 °C to +125 °C, the AD8429 is suitable for precision data acquisition applications.

The operational amplifier used is the OPA2810 from Texas Instruments, a high-performance dual-channel FET input operational amplifier with a gain bandwidth product of 70 MHz, a slew rate of 192 V/μs, and low input voltage noise. It supports a wide supply voltage range of 4.75 V to 27 V, and rail-to-rail input/output. Known for its low power consumption, low input offset current, and high linear output current, the OPA2810 excels in data acquisition and signal processing applications, and can operate stably over an industrial temperature range of −40 °C to +125 °C.

The programmable gain amplifier employed is the LTC6912-1 from Analog Devices, a dual-channel, low-noise, digitally programmable gain amplifier with independent gain control, a 3-wire SPI interface, rail-to-rail input/output, and low power consumption. It offers a flexible gain range up to 85 dB, provides a maximum offset voltage of 2 mV within the temperature range of −40 °C to 85 °C, and achieves a channel-to-channel gain matching of up to 0.1 dB.

## 3. Software Design

### 3.1. FPGA Software Design of the Main Control Board

The main control board is meticulously designed to integrate key components, such as a Digital Signal Processor (DSP), two Static Random-Access Memories (SRAMs), and an FPGA. The DSP is tasked with processing data from the telemetry circuit and interfacing with the main control FPGA via the External Memory Interface (EMIF) bus. The SRAMs serve primarily as data storage units. The FPGA’s programming design encompasses several distinct modules: EMIF bus interface module, command parsing module, control node module, data parsing module, and Universal Asynchronous Receiver/Transmitter (UART) communication module. The EMIF bus interface module facilitates data communication between the FPGA and the DSP. The command parsing module retrieves data from the EMIF bus interface, decodes it, and formats commands for transmission to the communication module, which then dispatches these commands to the acquisition and excitation circuits. The control node module ascertains which acquisition boards should be activated based on the received commands, and sends data request commands to each board in sequence. This module ensures precise data transmission, and systematically organizes the data collected from various acquisition boards. The data parsing module, in turn, directs the aggregated data to the FIFO buffer. These modules collectively enable the system to achieve real-time data acquisition and processing across multiple nodes, with the processed data being forwarded to the telemetry circuit. Figure 3 illustrates the FPGA program design block diagram of the main control board, highlighting the FPGA’s role in complex multi-node data acquisition and processing. Efficient communication between modules through bus interfaces is a key feature of this design, ensuring high efficiency and reliability in data transmission. This design meets stringent requirements for real-time performance and processing capability, making it well-suited for a variety of high-performance, high-throughput data acquisition applications.

Figure 4 presents the flowchart for the FPGA program design of the main control board. The program initiates by waiting to receive commands from the telemetry circuit. Upon command reception, the system first validates the command’s authenticity. If the command passes the verification, it is formatted and disseminated accordingly. The system then anticipates responses from each acquisition board. Should a response from any acquisition board be absent, the FPGA on the main control board activates a point-to-point retransmission protocol. Following the telemetry circuit’s directive, the FPGA dispatches a start acquisition command to the enabled acquisition boards. Upon completion of the acquisition process, the FPGA issues a read data command to each acquisition node in the sequence, and the collected data from each board is deposited into the FIFO buffer.

To prevent errors in the command dissemination process due to the substantial data volume on the RS485 bus, the main control board FPGA implements a CRC (Cyclic Redundancy Check) verification for the broadcast commands that need to be sent. The CRC check ensures data integrity by detecting any changes that may occur during transmission. The verification result is appended at the end of the broadcast command. The CRC formula used is as follows:(1)$$G\left(X\right)=X^{16}+X^{12}+X^{5}+1$$

To prevent system instability caused by external factors, the main control board FPGA is equipped with a maximum return time of 50 ms for data transmission from each acquisition board. This safeguard ensures reliability and stability during the data communication process. Should the maximum return time be exceeded, the main control board automatically switches to receive data from the next acquisition board, filling the data from the non-responsive board with zeros. This design mechanism allows the system to promptly identify and determine the faulty circuit board in cases of missing data transmission or board failure, thereby ensuring continuous operation and integrity of the entire system.

### 3.2. FPGA Software Design for the Acquisition Board

Figure 5 delineates the FPGA software architecture for the acquisition board, which is primarily composed of the data acquisition module, Serial Peripheral Interface (SPI) communication module, command parsing module, and Universal Asynchronous Receiver/Transmitter (UART) communication module. The UART communication module is tasked with receiving broadcast commands from the main control board and transmitting the collected acquisition data back to it. The command parsing module deciphers these broadcast commands and forwards the sampling parameters to the data acquisition module, while the gain parameters, once decoded, are relayed to the SPI communication module. The data acquisition module configures the Analog-to-Digital Converter (ADC) chip on the board and captures subsurface acoustic signals, as per the parsed command details, including sampling interval, rate, and depth. The gathered data is subsequently stored in the FIFO buffer. The SPI communication module dispatches the gain code to the Programmable Gain Amplifier (PGA) via the SPI bus. The FPGA software on the acquisition board facilitates real-time data acquisition and ensures the system operates flexibly and efficiently by dynamically adjusting acquisition parameters in response to varying commands.

Figure 6 outlines the flowchart for the FPGA program design of the acquisition board. The program commences operation by awaiting the reception of a broadcast command from the main control board. Once a command is received and its authenticity is successfully verified, the system parses the necessary acquisition parameters and proceeds to initialize the ADC. In this system, the AD7380 analog-to-digital converter is selected for its acquisition capabilities, boasting a 16-bit resolution and a maximum sampling rate of 4 Mega Samples Per Second (MSPS). Initially, the AD7380’s operational mode and sampling rate, along with other pertinent parameters, are configured. Subsequently, the conversion process is initiated, and the data collected is stored in the FIFO buffer in byte format, ready to be retrieved upon the main control board issuing a data read command.

## 4. Experimental Verification

Figure 7 presents the FPGA simulation for the initialization configuration of the ADC. When the chip select signal (CS) is low, the AD7380 is enabled to initiate communication with the FPGA. A high level of CS signifies the termination of the current communication session. During each low phase of CS, the serial clock signal (SCLK) produces a series of pulse signals. On the rising edge of the SCLK, the serial data input signal (SDI) commences the transmission of initialization commands. Upon completion of the initialization, CS returns to a high state, SCLK remains high, and SDI is set low.

Figure 8 depicts the EMIF bus timing simulation diagram, showcasing the interaction between the DSP and the FPGA, with the DSP assuming the role of command initiator. The DSP commences by writing pertinent data to specific addresses on the bus. Address 0 is designated as the bus test address, serving the purpose of conducting preliminary bus diagnostics to ascertain the proper functionality of bus communication prior to initiating any data exchange. Address 3 is earmarked as the gain parameter configuration address, which facilitates the issuance of gain control commands across all acquisition channels. Address 5 is assigned as the transmission parameter configuration address, dedicated to the dispatch of commands pertaining to the excitation board’s pulse emission, emission delay, and emission mode. Address 8 is identified as the acquisition parameter configuration address, which is instrumental in broadcasting commands that dictate the sampling depth, interval, and delay for all acquisition boards. Address 9 signifies the enable acquisition board address, determining the operational status of the designated acquisition boards based on the commands input to this address. Lastly, Address c is designated as the start transmission address, playing a crucial role in achieving synchronization across all acquisition boards and the excitation board.

Upon the DSP issuing commands via the EMIF bus, the main control board’s FPGA parses all commands and converts them into RS485 format, as depicted in Figure 9. The FPGA, after interpreting the bus commands, prepends a protocol header byte (16’hAE86) at the commencement of the command, followed by acquisition board parameter commands, enables information for the acquisition boards, gains parameter information for the acquisition boards, and transmits board command parameters, totaling 18 bytes. Subsequently, it performs a CRC calculation on the protocol header along with the command parameters, encompassing a total of 20 bytes, and appends the resulting checksum to the 21st byte. The main control board then dispatches the commands to all acquisition boards and transmission boards via the RS485 bus, awaiting acknowledgment of receipt.

The multi-node data acquisition system’s overall physical configuration is depicted in Figure 10. Comprising one main control board, eight acquisition boards, one communication board, and upper computer software, the system’s architecture is robust. The main control board and the acquisition boards are linked via an RS485 bus, facilitating data transfer. Each acquisition board relays the collected data to the FPGA on the main control board, which then forwards it to the DSP via the EMIF bus, as detailed in Figure 11. The main control board routes the collected data to the communication board over the RS485 bus, which in turn uploads the data to the upper computer software using a network cable, concurrently generating waveforms.

In the experimental phase, a sinusoidal excitation signal with a frequency of 4 MHz and an amplitude of 100 mV was employed. With a sampling interval set to 8 us, the waveform generated by the DSP indicated that one complete cycle of the sinusoid comprised 200 sampling points, aligning with the characteristics of the excitation source. The testing confirmed the ability of the acquisition system to accurately collect the signal, and to render it fully and accurately.

To ascertain the system’s signal collection stability and integrity, a 10 kHz sine wave signal was introduced at the input of each acquisition board. Collection commands were issued from the upper computer. As depicted in Figure 12, the sine wave signal is fully rendered on the graphical interface. As collection commands were continuously dispatched while the amplitude and frequency of the signal source were varied, the waveforms were accurately and consistently displayed on the upper computer software.

In order to evaluate the system’s throughput, the collection depth was configured to 2048, with all eight acquisition boards active, and a sampling interval of 250 ns (equivalent to a 4 MSPS sampling rate for the ADC). Under these parameters, the data volume collected per iteration was calculated to be 2048 × 8 × 8 × 2 = 262,144 bytes. Following a stress test where collection commands were repeatedly issued over a 20-min period, all waveforms were displayed without issue. This test confirms that the system satisfies the high data acquisition volume and stable transmission performance necessary for logging-while-drilling acoustic applications.

## 5. Conclusions

This study presents the research and design of a multi-node data acquisition system for logging-while-drilling acoustic logging, based on FPGA technology. The system enhances flexibility and efficiency, achieving higher data acquisition rates and more comprehensive subsurface information gathering. The experimental results show that the waveform signal can be accurately collected and adapted to the multi-channel acquisition system.

The use of an RS485 bus for connecting multiple acquisition boards to the main control board offers advantages such as extended communication distance and strong anti-interference capabilities. The acquisition boards can be distributed at different locations to simultaneously collect subsurface data. This interconnection enables centralized data management, thereby improving the overall efficiency and reliability of the system.

The system exhibits good scalability, allowing for time-division control of the acquisition boards via the FPGA on the main control board. This flexibility enables the addition or removal of acquisition boards based on different application scenarios, facilitating the acquisition of more comprehensive subsurface information.

Logging-while-drilling acoustic logging requires high-precision data to accurately reflect the structure and characteristics of underground rock. High precision implies larger data volumes; thus, this system primarily utilizes FPGA chips for data acquisition and processing, benefiting from strong parallel computing capabilities and fast data processing speeds. This meets the demands of logging-while-drilling acoustic logging for data processing speed, real-time performance, flexibility, and reliability.

Experimental testing has validated that the system can achieve complete transmission of waveform signals, functioning normally under various acquisition parameters. The transmission performance is excellent, fulfilling the requirements for acoustic signal acquisition in subsurface applications.

## Figures and Tables

**Figure 1 sensors-25-00808-f001:**
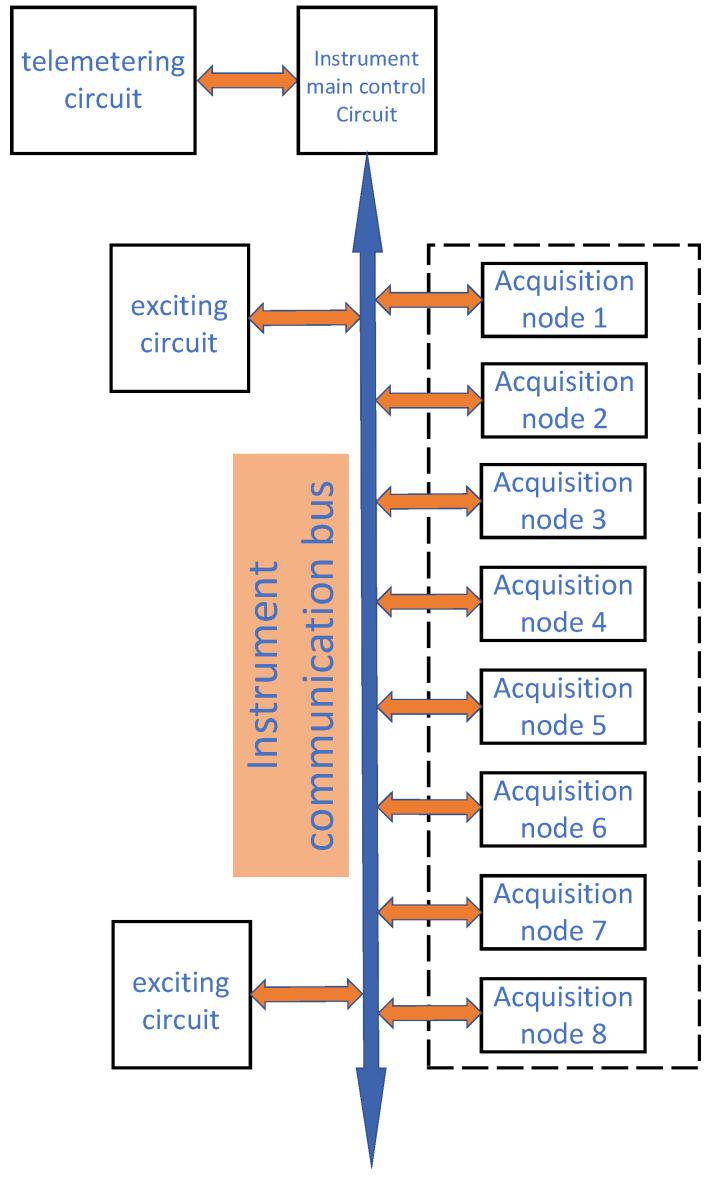
A schematic diagram of the electronic system composition for logging-while-drilling acoustic logging instrument.

**Figure 2 sensors-25-00808-f002:**
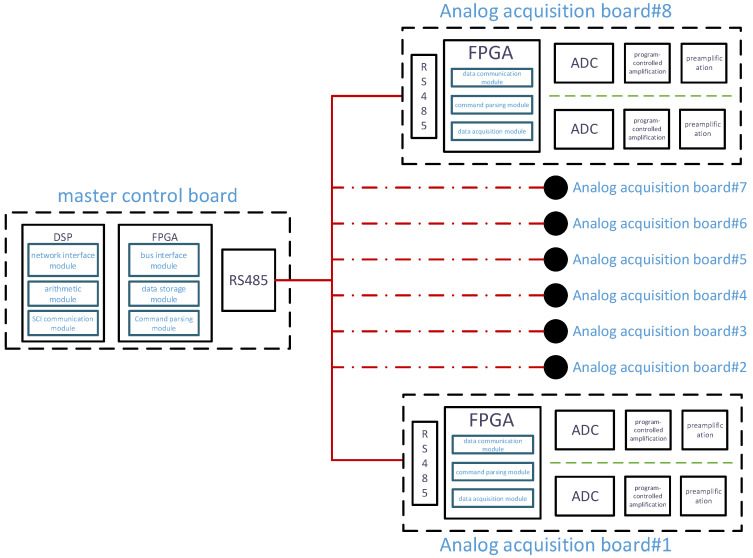
A block diagram of the overall architecture for the multi-node data acquisition system of logging-while-drilling acoustic logging instrument.

**Figure 3 sensors-25-00808-f003:**
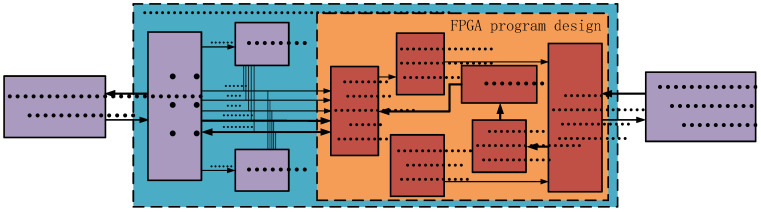
Block diagram of the main control board FPGA program design.

**Figure 4 sensors-25-00808-f004:**
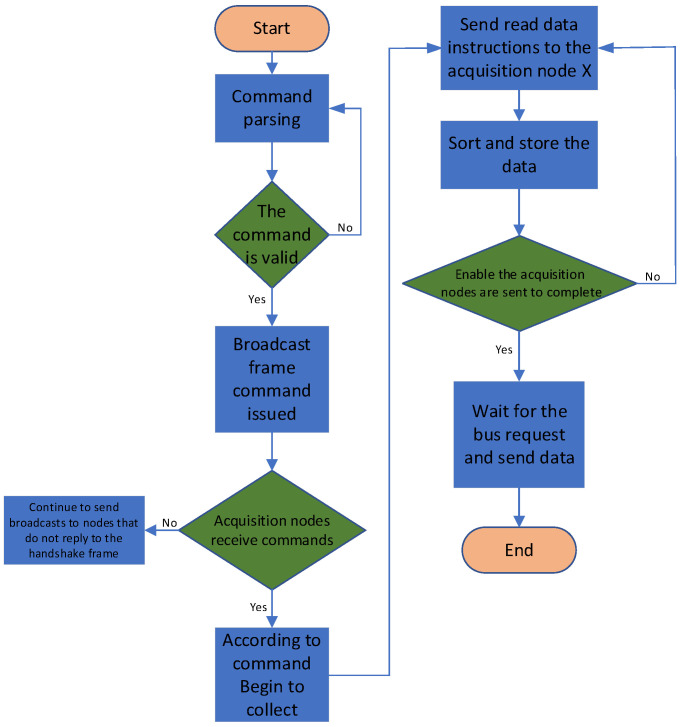
Flowchart of the FPGA program design for the main control board.

**Figure 5 sensors-25-00808-f005:**
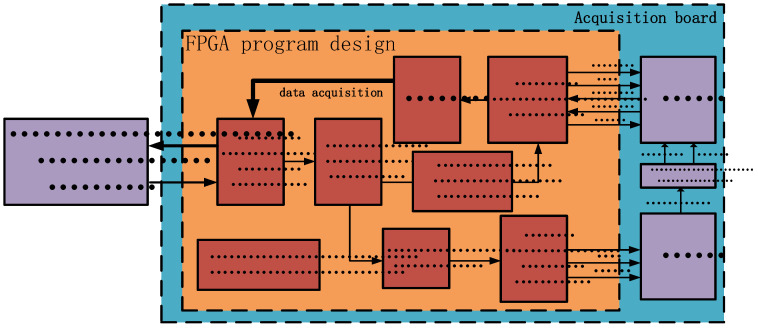
Block diagram of the FPGA program design for the data acquisition board.

**Figure 6 sensors-25-00808-f006:**
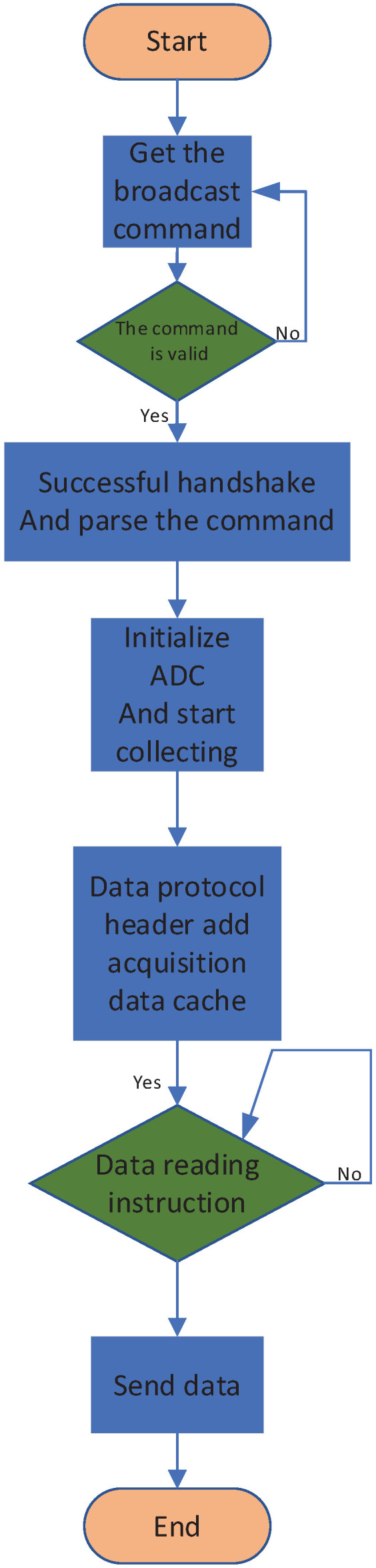
Flowchart of the FPGA program design for the acquisition board.

**Figure 7 sensors-25-00808-f007:**
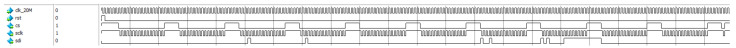
Timing diagram for ADC initialization.

**Figure 8 sensors-25-00808-f008:**
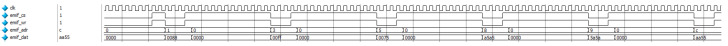
EMIF bus timing simulation diagram.

**Figure 9 sensors-25-00808-f009:**
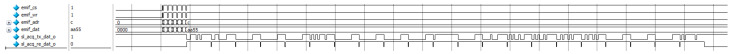
Broadcast command timing simulation diagram.

**Figure 10 sensors-25-00808-f010:**
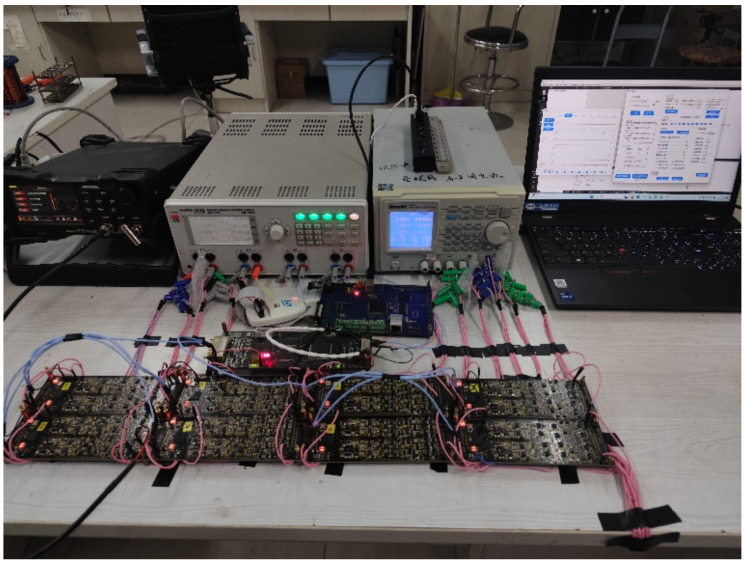
Figure of the overall physical setup for experimental testing.

**Figure 11 sensors-25-00808-f011:**
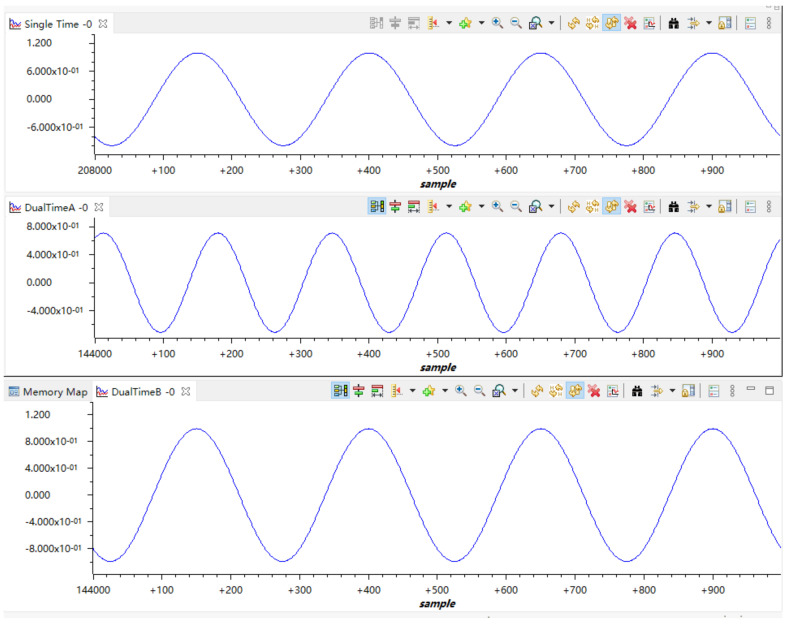
Acquisition of sinusoidal waves with different frequencies and amplitudes.

**Figure 12 sensors-25-00808-f012:**
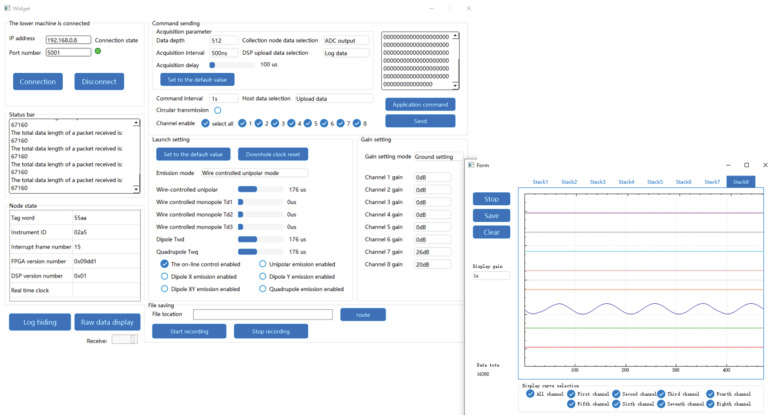
Upper computer software displays waveform graph.

## Data Availability

The original data presented in the study are included in the article.

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
