# Peer review of "Design of a Multi-Node Data Acquisition System for Logging-While-Drilling Acoustic Logging Instruments Based on FPGA"

_sensors, 2025, doi:10.3390/s25030808_

Round 1
Reviewer 1 Report
Comments and Suggestions for Authors
This paper designed a FPGA-based logging-while-drilling (LWD) multi-node data acquisition system, which increases sampling density and data accuracy related to current LWD acquisition systems. However, there are some issues should be addressed, which present as follows.
1. In application, the acquired signal come from acoustic transducers, so the authors should provide the experiment results of collecting signals by acoustic transducers.
2. In Figure 1, telemetering circuit is one important component of the data acquisition system. The author should elaborate on the telemetry method utilized for information transmission in acoustic logging while drilling, including a detailed introduction of the relevant telemetry subsystem's function, circuit design, and experimental validation.
3. The master control board communicate with analog acquisition board by RS485, their communication mode is half duplex in fact, therefore, the main control board can only communicate with one of the acquisition boards at any time. Hence, the authors should provide a detailed introduction of the communication mechanism and protocol.
4. Between lines of 164 and 166, it is mentioned that the bandwidth of AD8429 is 15MHz, and its bandwidth is 1.2MHz at a gain of 100, the gain bandwidth product is not equal to its bandwidth, please give a reasonable explanation.
5. In Figure 5, the relationship among PGA, ADC and input signal in not clear, the authors should improve it.
6. In this design, the authors present three operational amplifiers, but not introduce the elctronic circuits of these operational amplifiers.
7. In Figure 1, the exciting circuit is very important in acoustic logging, but the exciting circuit and its control method are not presented.
8. The authors should provide one section for introducing the hardware.
9. The width of Figures 7, 8 and 9 is much wide than text, it should be improved.
10. The introduction needs improvement to illustrate the importance and necessity of the author's work.
There is no innovation to be found in this article. I do not recommend this article for publication.
Comments on the Quality of English Language
The article's English should be improved for better readability and there are some typos in the manuscript.
Author Response
RE: sensors-3413171
We are very grateful to your and the reviewers’ critical comments and thoughtful suggestions. Based on these comments and suggestions, we have made careful modification on the original manuscript. All changes made to the text are in red in the revised manuscript so that they may be easily identified. Some of your questions were answered below.
Comments 1: In application, the acquired signal come from acoustic transducers, so the authors should provide the experiment results of collecting signals by acoustic transducers.
Response 1: Thank you very much for your opinion. If in the actual downhole application process, it must be to collect the signal of the acoustic transducer. However, at present, the paper is designed as a scientific prototype, which only supports method verification, and has not yet done specific work on the acquisition of the acoustic transducer. This is the focus of our future consideration. Thank you very much for your advice.
Comments 2: In Figure 1, telemetering circuit is one important component of the data acquisition system. The author should elaborate on the telemetry method utilized for information transmission in acoustic logging while drilling, including a detailed introduction of the relevant telemetry subsystem's function, circuit design, and experimental validation.
Response 2: Thank you very much for your suggestion, I will make the following explanation. The remote transmission circuit is a very important part of the logging instrument. Its main function is to issue the command of the logging vehicle and upload the collected data. This paper mainly studies the multi-node acquisition system, which is not working with the upper remote transmission circuit for the time being. This will be an important issue for us to consider in the future, thank you.
Comments 3: The master control board communicate with analog acquisition board by RS485, their communication mode is half duplex in fact, therefore, the main control board can only communicate with one of the acquisition boards at any time. Hence, the authors should provide a detailed introduction of the communication mechanism and protocol.
Response 3: Thank you very much for your advice. The communication between the main control board and the acquisition board is half-duplex, so each time the main control board is required to actively poll the acquisition board. Different acquisition boards treat commands differently through different communication addresses, which can complete the communication between multiple acquisition boards of a main control board.
Comments 4: Between lines of 164 and 166, it is mentioned that the bandwidth of AD8429 is 15MHz, and its bandwidth is 1.2MHz at a gain of 100, the gain bandwidth product is not equal to its bandwidth, please give a reasonable explanation.
Response 4: Thank you very much for pointing out the problem, I have found my wrong writing and to delete.
Comments 5: In Figure 5, the relationship among PGA, ADC and input signal in not clear, the authors should improve it.
Response 5: Thank you for your suggestion. I have added the relationship between PGA and ADC in Figure 5 and marked it in the figure.
Comments 6: In this design, the authors present three operational amplifiers, but not introduce the elctronic circuits of these operational amplifiers.
Response 6: Thank you very much for pointing out the problem, I will make the following explanation. Because this paper focuses on the research of multi-node acquisition system, and focuses on the research and development based on FPGA. Therefore, the operational amplifier of the previous stage is not introduced in detail. This will be our future work to increase.
Comments 7: In Figure 1, the exciting circuit is very important in acoustic logging, but the exciting circuit and its control method are not presented.
Response 7: Thanks very much ! Firstly, the excitation circuit is very important in acoustic logging, which is mainly used to generate pulses and stimulate the characteristics of transducers. However, this paper focuses on the acquisition circuit system, so there is not much introduction to the excitation circuit.
Comments 8: The authors should provide one section for introducing the hardware.
Response 8: Thank you very much for your opinion. First of all, this paper has a description of the whole system and hardware design in the introduction part, and introduces the overall hardware design scheme and related chip selection. From the beginning of Chapter 2, the software of the system is introduced.
Comments 9: The width of Figures 7, 8 and 9 is much wide than text, it should be improved.
Response 9: Thank you for your suggestion. I have modified the figure and marked it in red in the text.
Comments 10: The introduction needs improvement to illustrate the importance and necessity of the author's work.
Response 10: Thank you very much for your comments, I have revised the introduction and increased the importance and necessity of the work.
Once again, we acknowledge your comments and constructive suggestions very much, which are valuable in improving the quality of our manuscript.
Kind regards
Sincerely yours
Qin Zhenyu

Reviewer 2 Report
Comments and Suggestions for Authors
Acoustic logging while drilling instruments play an important role in petroleum exploration, and this manuscript introduces the design of a multi-node data acquisition system for acoustic LWD based on FPGA technology. Although the use of FPGA high-speed processing, RS485 multi-node communication and other contents for logging circuit work are common technologies, the paper has reference value for the actual circuit and test work.
My comments is as follows:
1. The manuscript is about downhole measurement circuits, and the author should explain the considerations for designing the downhole circuit, such as temperature characteristics, noise handling, reliability, etc. For example, the authors list some IC properties from it datasheet, most of them are up to 125°C, but some are only up to 85°C. Why use a chip with such a temperature range?
2. The author wrote two FPGAs in the paper, one is AMD's XC7K325T, and the other is Intel's (Altera's) 10M16SCU169I7G, what is the relationship between these two FPGAs needs to be explained clearly. The authors chose a dual core DSP processor TMS320F28377D, and there is no description in the paper as to why they chose a dual core processor, how the two cores were used or coordinated to work together, or was only one of the cores used?
Author Response
RE: sensors-3413171
We are very grateful to your and the reviewers’ critical comments and thoughtful suggestions. Based on these comments and suggestions, we have made careful modification on the original manuscript. All changes made to the text are in red in the revised manuscript so that they may be easily identified. Some of your questions were answered below.
Comments 1: The manuscript is about downhole measurement circuits, and the author should explain the considerations for designing the downhole circuit, such as temperature characteristics, noise handling, reliability, etc. For example, the authors list some IC properties from it datasheet, most of them are up to 125°C, but some are only up to 85°C. Why use a chip with such a temperature range?
Response 1: The downhole measurement circuit generally requires high temperature, but our current design is a scientific prototype, which is only tested in a standard well, and the selected chips meet the requirements of the measurement environment. Thank you for your comments.
Comments 2: The author wrote two FPGAs in the paper, one is AMD's XC7K325T, and the other is Intel's (Altera's) 10M16SCU169I7G, what is the relationship between these two FPGAs needs to be explained clearly. The authors chose a dual core DSP processor TMS320F28377D, and there is no description in the paper as to why they chose a dual core processor, how the two cores were used or coordinated to work together, or was only one of the cores used?
Response 2: Thank you for your opinion, I made the following explanation, the design uses two fpga, which XC7K325T large amount of resources, is used for the main control circuit to store all the acquisition board data ; the design of 10M16SCU169I7G is simple, and only needs 3.3V power supply, which is used to control the acquisition circuit and receive commands from the main control board. At present, I use a dual-core DSP processor TMS320F28377D, and now only use one core to issue commands and upload data. In the future, the real-time processing of downhole data is planned to be carried out in DSP, so the dual-core processor is used, and the other core is later used for real-time processing of downhole data.
Once again, we acknowledge your comments and constructive suggestions very much, which are valuable in improving the quality of our manuscript.
Kind regards
Sincerely yours
Qin Zhenyu

Reviewer 3 Report
Comments and Suggestions for Authors
The paper proposes and implements a FPGA-based Multi-Node Data Acquisition System for Logging-While-Drilling Acoustic Logging Tools with a clear design concept and feasible implementation method, which has certain practicality and academic value. The design is not limited to the traditional architecture and highlights the advantages of FPGA, which is innovative and scalable. The paper analyzes the testing methods for the control timing and overall functionality, and provides test results and conclusions.
The paper needs to be revised and improved, it is recommended that the authors make the following modifications and improvements:
1.The language in the text can be further optimized and the expression should be clear and explicit;
2.In this paper, the importance of multi-node acquisition system in acoustic logging while drilling tool is not highlighted. It is suggested to analyze the important role of multi-node acquisition system in acoustic logging while drilling tool.
3.The research methods used in the paper are not sufficiently summarized in the abstract.
4. Are the accompanying figures in line with the formatting requirements of the paper? Please ensure that the size, color, and font of the figures meet the formatting requirements of the paper to better present the overall style and visual effect of the paper.
5. There are some specific points that need to be checked regarding the format, accuracy, and journal requirements of references in the paper. Please carefully check to ensure that all references have correct format and are accurately cited.
6. Further refine the conclusions and highlight summaries of innovative content in the paper.
In summary, based on the reviewer's opinion, this paper has certain academic and practical value, and it is recommended for acceptance after revision.
Author Response
RE: sensors-3413171
We are very grateful to your and the reviewers’ critical comments and thoughtful suggestions. Based on these comments and suggestions, we have made careful modification on the original manuscript. All changes made to the text are in red in the revised manuscript so that they may be easily identified. Some of your questions were answered below.
Comments 1: The language in the text can be further optimized and the expression should be clear and explicit;
Response 1: Thanks for your suggestion. We have tried our best to optimize the language in the revised manuscript. We have used technical terminology wherever possible and made modifications to some of the words.
Comments 2: In this paper, the importance of multi-node acquisition system in acoustic logging while drilling tool is not highlighted. It is suggested to analyze the important role of multi-node acquisition system in acoustic logging while drilling tool.
Response 2: Thank you for your comments, I very much agree with your comments, I have added in the introduction to the multi-node acquisition system in the important role of acoustic logging while drilling instruments, analyzes the important role of multi-node acquisition system.
Comments 3: The research methods used in the paper are not sufficiently summarized in the abstract.
Response 3: Thank you very much for your opinion. I have added a summary of the research methods designed in this paper according to your opinion. Thank you again !
Comments 4: Are the accompanying figures in line with the formatting requirements of the paper? Please ensure that the size, color, and font of the figures meet the formatting requirements of the paper to better present the overall style and visual effect of the paper.
Response 4: Thank you for your suggestions. We have enhanced the clarity of the images as much as possible and increased the font size within the images while ensuring no distortion occurs. Additionally, we have used bolding in certain areas to improve visual presentation. After review, these changes meet the requirements of the manuscript format. The revised sections are marked in red font. We greatly appreciate your attention to detail.
Comments 5: There are some specific points that need to be checked regarding the format, accuracy, and journal requirements of references in the paper. Please carefully check to ensure that all references have correct format and are accurately cited.
Response 5: Thank you for your suggestions. We have meticulously reviewed the format of the references and made necessary revisions. The specific modifications are highlighted in red text within the manuscript.
Comments 6: Further refine the conclusions and highlight summaries of innovative content in the paper.
Response 6: We consider this to be a highly constructive suggestion. Based on your advice, we have refined the conclusions to make them more succinct. The analysis now begins with a summary of the paper's innovative aspects and includes a summary of the system's stability and feasibility. Specific modifications are highlighted in red text.
Once again, we acknowledge your comments and constructive suggestions very much, which are valuable in improving the quality of our manuscript.
Kind regards
Sincerely yours
Qin Zhenyu

Round 2
Reviewer 1 Report
Comments and Suggestions for Authors
The revised manuscript still fails to provide a novel contribution to the methodology of acoustic logging while drilling as well as the instrumentation. Therefore, I do not recommend this paper for publication in Sensors.
Author Response
RE: sensors-3413171:
Thank you for your suggestions. We have modified the manuscript as much as possible. The author believes that the multi-node acquisition system is a very important part of the acoustic logging while drilling. Due to the large amount of acoustic logging data while drilling and the detection of more detailed formation information, multiple acquisition boards are required to collect formations in different directions. The system designed in this paper can adapt to the current needs.
Once again, we acknowledge your comments and constructive suggestions very much, which are valuable in improving the quality of our manuscript.
Kind regards
Sincerely yours
Qin Zhenyu